# Experimental and Theoretical Study on Crown Ether-Appended-Fe(III) Porphyrin Complexes and Catalytic Oxidation Cyclohexene with O_2_

**DOI:** 10.3390/molecules28083452

**Published:** 2023-04-13

**Authors:** Xiaodong Li, Ailing Feng, Yanqing Zu, Peitao Liu, Fengbo Han

**Affiliations:** Institute of Physics & Optoelectronics Technology, Baoji University of Arts and Sciences, Baoji 721016, China; lixd@bjwlxy.edu.cn (X.L.); zuyanqing74@bjwlxy.edu.cn (Y.Z.);

**Keywords:** crown ether-appended Fe(III) porphyrin complexes, catalytic oxidation, cyclohexene, DFT analysis, reaction mechanism

## Abstract

Modifying non-precious metal porphyrins at the *meso*-position is sufficient to further improve the ability to activate O_2_ and the selectivity of the corresponding redox products. In this study, a crown ether-appended Fe(III) porphyrin complex (FeTC_4_PCl) was formed by replacing Fe(III) porphyrin (FeTPPCl) at the *meso*-position. The reactions of FeTPPCl and FeTC_4_PCl catalysed by O_2_ oxidation of cyclohexene under different conditions were studied, and three main products, 2-cyclohexen-1-ol (**1**), 2-cyclohexen-1-one (**2**), and 7-oxabicyclo[4.1.0]heptane (**3**), were obtained. The effects of reaction temperature, reaction time, and the addition of axial coordination compounds on the reactions were investigated. The conversion of cyclohexene reached 94% at 70 °C after 12 h, and the selectivity toward product **1** was 73%. The geometrical structure optimization, molecular orbital energy level analysis, atomic charge, spin density, and density of orbital states analysis of FeTPPCl, FeTC_4_PCl, as well as the oxygenated complexes (Fe-O_2_)TCPPCl and (Fe-O_2_)TC_4_PCl formed after adsorption of O_2,_ were carried out using the DFT method. The results of thermodynamic quantity variation with reaction temperature and Gibbs free energy variation were also analysed. Finally, based on experimental and theoretical analysis, the mechanism of the cyclohexene oxidation reaction with FeTC_4_PCl as a catalyst and O_2_ as an oxidant was deduced, and the reaction mechanism was obtained as a free radical chain reaction process.

## 1. Introduction

Metalloporphyrin complexes exhibit exceptional catalytic properties for various chemical transformations, including O–O bond heterolysis [1,2,3,4], H–H bond formation [5,6], O_2_ adsorption [4,7,8], and catalytic oxidation [9,10,11,12] due to high activity, stability, clear molecular structure, and the ability to mimic the cytochrome enzyme [13,14]. Previous studies have shown that changing the *meso*-position substituents of metalloporphyrins to modulate their redox properties can alter metal ions’ activities, stability, and selectivity in catalytic processes [15,16,17,18,19,20,21]. Strong electron-absorbing *meso*-position substituents can make the metalloporphyrins easy to reduce, while electron-donating substituents can increase the electron-cloud density on the metal ion, enhancing its binding and electron transfer ability with O_2_ [22,23]. Therefore, fine-tuning metal ions’ electronic structure by changing the porphyrin ring’s *meso*-position substituents is vital for further improving their O_2_-carrying ability [7,20,24]. Additionally, the planar structure of metalloporphyrins provides a good coordination environment for modifying axial ligands [2,17,18,25,26,27]. The unoccupied d orbitals of the active metal centre can receive electrons provided by the ligand, increasing the electron density of the metal ion active centre or forming an electron-deficient structure that acts as a Lewis acid centre at the axial vacant coordination site, which is conducive to O_2_ adsorption.

In this study, the pre-synthesized *meso*-5, 10, 15, and 20-*tetra*-(15-crown-5-benzo-3-yl-sulfonyloxophen-4-yl) Fe(III) porphyrin complex (FeTC_4_PCl) was used as the catalyst [28] (Figure 1a). The catalytic oxidation of cyclohexene was carried out by using O_2_ as an oxidant under mild conditions without any auxiliary to verify the effect of the substitution of the *meso*-position with the 15-crown-5-benzo-3-sulfonyl group on the catalytic ability of the Fe(III) porphyrin, and in comparison with that of Fe(III) porphyrin complex (FeTPPCl, Figure 1b) without this group, the results demonstrate that the FeTC_4_PCl has higher product selectivity and substrate conversion as well as exhibits the ability to shorten the reaction induction period. This change indicates that the *meso*-substituent can effectively influence the catalytic activity of Fe(III). In contrast, the oxidation selectivity for the product 2-cyclohexene-1-ol was significantly higher than that of the FeTPPCl complex. The changes in catalytic oxidation at different temperatures and reaction times as well as the addition of axial ligand (pyridine) were also examined.

To further analyse the effect of crown ether ring substituents in FeTC_4_PCl on the properties of Fe(III) porphyrins, the geometric configurations and corresponding orbital energy level distributions of the FeTC_4_PCl and FeTPPCl complexes were calculated separately using the density functional theory (DFT), and the changes of the thermodynamic quantities of FeTC_4_PCl at different temperatures were analysed separately. On this basis, the stable geometries of the oxygenated complexes formed after the adsorption of O_2_ by the FeTC_4_PCl and FeTPPCl complexes were also constructed. The geometric configurations and corresponding properties such as chemical bond lengths and density-of-states were calculated. Finally, the reaction mechanism of the catalytic oxidation of cyclohexene by FeTC_4_PCl in the presence of O_2_ was deduced by combining the above analysis results, and the reaction process was obtained as a free radical chain reaction-based reaction process.

## 2. Experimental Section

### 2.1. Materials and Catalytic Property Studies

All chemicals were obtained commercially and were further purified. High-purity O_2_ gas was purchased from a reagent supplier and purified before use. Catalysts FeTPPCl and FeTC_4_PCl were prepared according to the previous synthesis method of our group [28] and the corresponding structures of the complexes are shown in Figure 1.

The catalytic activity of FeTC_4_PCl and FeTPPCl complexes was evaluated for the oxidation of cyclohexene with O_2_ following the literature procedure [28,29]. FeTC_4_PCl (2.0 mg, 0.00096 mmol) and cyclohexene (2.0 mL, 19.76 mmol) were added to a round bottom flask. The flask was evacuated and charged with anhydrous oxygen. The reaction mixture was heated in an oil bath under stirring, and the corresponding experimental reaction apparatus is shown in Appendix A. The consumption of oxygen was measured, and the corresponding products were analysed by gas chromatography using an Agilent 19091S-433 HP-5MS phenyl methyl siloxane column with a 30.0 m × 0.25 μm column at a temperature range of 70–230 °C (10 °C/min), Inj. 230 °C (Dec. 230 °C). All products were analysed by mass spectrometry as well as compared to the standard masses of organic compounds and their fragmentation patterns. The main oxidation products were 2-cyclohexen-1-ol (**1**), 2-cyclohexen-1-one (**2**), and 7-oxabicyclo[4.1.0]heptane (**3**) (Figure 1).

### 2.2. Computational Details

In this paper, all the geometry optimizations and single-point calculations were performed using the Gaussian 09 package [30]. The optimization for FeTC_4_PCl was carried out using three hybrid functionals, namely PBE0 [31], ωB97XD [32], and B3LYP [33], in conjunction with the def2-SVP basis set [34] for geometry and the def2-TZVPP basis set for single-point calculations in the gas phase. The DFT-D dispersion correction proposed by Grimme [35] was added to the calculation process to remedy the deficiencies in the description of the electron-correlation interaction by the two general functionals, PBE0 and B3LYP. All initial geometries were in the sextet spin state, and the optimized structures discussed were characterized as local minima without imaginary frequencies. In order to select the most appropriate functional, the IR spectra of FeTC_4_PCl were calculated under the def2-SVP basis set to facilitate comparison with the experimental values. The structure of FeTC_4_PCl is presented in Appendix A, which corresponds to the calculated bond lengths and angles shown in Appendix A along with the experimental values. The computational results using ωB97XD/def2-SVP are in agreement with the experimental values. Therefore, the optimizations in this work were performed using the ωB97XD functional. Based on the optimised structure, the single-point energy was obtained at the ωB97XD/def2-TZVPP level. To further analyse the binding ability of FeTC_4_PCl and FeTPPCl complexes with O_2_ molecules and the geometric configuration of the oxygenated complex, and to understand the influence of *meso*-site substituent groups on the adsorption of O_2_ molecules, the geometric configurations of the oxygenated complex (Fe-O_2_)TPPCl and (Fe-O_2_)TC_4_PCl formed by the above two complexes were simulated and constructed. Their geometric optimization and analysis were carried out using ωB97XD/def2-SVP, and single-point calculation was carried out at the TZVPP level. The entire optimized structure was a local minimum without imaginary frequencies.

The electronic structure analyses were performed using the Multiwfn 3.8 (dev) code [36], and the isosurface maps of various orbitals and real space functions were plotted using Visual Molecular Dynamics (VMD) [37] software based on the files exported from Multiwfn. In addition, Shermo [38] analysis software was used to analyse thermodynamic quantities throughout the process of forming the oxygenated complex (Fe-O_2_)TPPCl and (Fe-O_2_)TC_4_PCl during the combination of FeTPPCl and FeTC_4_PCl with O_2_. The variation trends of internal energy (U), enthalpy (H), and Gibbs free energy (G) with increasing temperature was calculated.

## 3. Results and Discussion

### 3.1. Catalytic Property of Complexes

Table 1 summarizes the main results obtained under different reaction conditions. The results indicate that conversion was considerably enhanced when the reaction temperature was increased from 30 °C to 70 °C (entries 1–5). However, attempts to further enhance the conversion by increasing the reaction temperature were unsuccessful, and the conversion decreased from 94% to 91% compared to entry 5 (entry 6). These results suggest that the optimal reaction temperature for the oxidation of cyclohexene is 70 °C.

The effect of product selectivity at different reaction temperatures was also discussed. As shown in Table 1, the highest product selectivity for **1** was observed at 50 °C (entry 3) rather than 70 °C. It was concluded that product **1** could be a possible intermediate product [18]. Compared to the catalytic oxidation of cyclohexene by FeCl_3_, FeTPPCl complexes, and *meso*-5,10,15,20-*tetra*-(15-crown-5-benzo-3-yl-sulfonyloxophen-4-yl)porphyrin compound (TC_4_HPP), it is evident that the FeTC_4_PCl complex demonstrates superior catalytic performance for the mild oxidation of cyclohexene with O_2_ (entries 5, 7–9). Importantly, the conversion reached up to 94% after four crown ether rings were appended to the porphyrin framework (entry 5). This phenomenon also supports the conclusion that benzo-15-crown-5 possessed a larger electron-donating environment compared to the Fe(Ⅲ) porphyrin ring, as it favours that the O_2_ molecule approaches the coordination centre of the Fe(Ⅲ) complex and stabilises the Fe–O_2_ bond. Compared to the uncrowned analogue Fe (Ⅲ) porphyrin (FeTPPCl), the reaction could not take place without the active centre (entry 10), strongly implying that O_2_ could not be activated without the FeTC_4_PCl complex or other catalysts.

### 3.2. Effect of meso-Substituents

FeTPPCl and FeTC_4_PCl were used as catalysts, and O_2_ was used as the oxidant to investigate the conversion of cyclohexene and product selectivity over time, as shown in Figure 2 and Figure 3. Figure 2 depicts the catalytic oxidation process of cyclohexene using FeTC_4_PCl as the catalyst, utilizing O_2_ as the oxidizing agent in the absence of any additives. FeTC_4_PCl, a complex featuring a 15-crown-5-benzo-3-sulfonyl group, demonstrated remarkable efficiency in enhancing the conversion rate of cyclohexene, as evidenced by the red line in Figure 2. The conversion rate increased significantly and rapidly with increasing reactivity. For instance, after 1 h of reaction time, the conversion rate was 8%, and it reached 49% after 2 h. Furthermore, after a reaction time of 12 h, the conversion rate of cyclohexene increased to 93%. In contrast, the utilization of FeTPPCl (a catalyst lacking the 15-crown-5-benzo-3-sulfonyl group) for the O_2_ oxidation of cyclohexene under the same reaction conditions (represented by the blank line in Figure 2) resulted in a substrate conversion rate of 7% after 1 h, which is similar to that obtained using FeTC_4_PCl. However, after 3 and 12 h of reaction time, the substrate conversion rates were 35% and 76%, respectively. Based on a comparison of the catalytic effects of the two catalysts, it can be concluded that FeTC_4_PCl exhibits superior catalytic activity. Furthermore, both catalysts significantly affected the selectivity toward the oxidation product of cyclohexene. Figure 3 shows that FeTC_4_PCl resulted in 70% and 53% selectivity for product **1** after 7 and 12 h, respectively, while FeTPPCl showed a lower selectivity of 56% and 40%, respectively. The effects of both catalysts on the selectivity of products **2** and **3** could have been more pronounced, which is possibly due to their low contents in the reaction mixture. Overall, the results demonstrate that FeTC_4_PCl with 15-crown-5-benzo-3-sulfonyl group substitution is a superior catalyst for conversion and selectivity compared to FeTPPCl.

### 3.3. Effect of Reaction Temperature

The oxidation of cyclohexene using FeTC_4_PCl as the catalyst showed the best catalytic activity. Therefore, the catalytic oxidation of cyclohexene using the FeTC_4_PCl catalyst at various temperatures was studied in further detail.

The catalytic activity of FeTC_4_PCl for the oxidation of cyclohexene by activated O_2_ was investigated in the temperature range of 30–77 °C at 10 °C increments. The conversion, turnover numbers (TON), and product selectivity of cyclohexene oxidation at various temperatures are listed in Table 1. The conversion of cyclohexene increased from 24% to 94% with a gradual increase in temperature from 30 °C to 70 °C (Entries 1 and 5 of Table 1), indicating that temperature had a noticeable effect on increasing the catalytic activity of FeTC_4_PCl. With a further increase in temperature to 77 °C, the conversion of cyclohexene decreased by 4% (Entry 5 vs. 6 of Table 1), indicating that the optimal reaction temperature was 70 °C. At temperatures above 70 °C, the conversion of cyclohexene decreased because of the high volatility of cyclohexene and the maximum oxygen consumption at 70 °C. These results were consistent with the maximum conversion characteristics of cyclohexene.

Table 1 demonstrates the significant influence of reaction temperature on product selectivity, specifically regarding **1** and **2**. An increase in temperature notably impacts their selectivity, as exemplified by the selectivity related to **1** reaching 53% and **2** decreasing to 12% (Entry 3 of Table 1) at 50 °C. These results suggest that raising the temperature enhances the conversion of cyclohexene, resulting in a relatively high yield of **1**. Furthermore, it is observed that **1** is produced primarily among the oxidation products of cyclohexene and remains stable at 50 °C. At 77 °C, the selectivity towards **1** increased to 73%, while the selectivity towards **2** remained virtually unchanged (Entry 5 of Table 1), indicating that a higher temperature facilitated the formation of **1**. The FeTC_4_PCl catalyst demonstrated relatively high catalytic activity at this temperature, thus promoting the selectivity regarding **1**. However, a decrease in total conversion and selectivity of **1** was observed at 77 °C (Entry 6 of Table 1). This phenomenon could be attributed to the possible decomposition of the FeTC_4_PCl catalyst and intermediate peroxides in the reaction, which may have been accelerated due to the higher temperature, leading to the decomposition of the peroxides by the metal porphyrin and the decomposition of the metal porphyrin by the peroxides [19]. Additionally, it was observed that at 70 °C, **1** rapidly oxidized to other products, indicating that 70 °C exceeded the optimum reaction temperature.

Among the oxidized products, **3** can be considered as an intermediate in the oxidation reaction [39,40,41]. The amount of **3** increased slightly with increasing reaction time, reaching only 3% after 6 h (Entry 5 of Table 1). Upon adding a stoichiometric amount of FeCl_3_ to the oxidation system, a lower conversion of cyclohexene (40%) was observed (Entry 8 of Table 1) and the selectivity of products **1** and **2** were relatively lower. This suggests that the 15-crown-5-benzo-3-sulfonyl group enhances the catalytic activity by promoting the activity of the Fe(III) centre. Cyclohexene was not oxidized in the absence of the catalyst (Entry 9 of Table 1), indicating the excellent catalytic effect of metal porphyrin catalysts.

### 3.4. Effect of Axial Ligand

Pyridine was chosen as the axial ligand to investigate its impact on the catalytic oxidation of cyclohexene using FeTC_4_PCl as the catalyst. Adding 5% pyridine to the oxidation system significantly affected the conversion and the selectivity of the oxidation products. The impact of pyridine on the catalytic oxidation reaction was illustrated in Figure 4. The results showed that pyridine inhibited cyclohexene’s catalytic oxidation using FeTC_4_PCl as the catalyst, leading to a reduction in cyclohexene conversion from 94% to 55%. Furthermore, the selectivity of **1** and **2** were lower than those without pyridine, while the selectivity of **3** was relatively higher. The axial coordination between the transition metal ion in FeTC_4_PCl and pyridine influenced the redox potential of the active metal centre, making it difficult for the active intermediate to coordinate with oxygen and form a high-valence state or transfer activated oxygen to cyclohexene, thereby inhibiting the oxidation reaction [17].

### 3.5. Optimized Structure and Bonding Properties

The optimized geometries of FeTPPCl and FeTC_4_PCl in the sextet ground spin state and that of (Fe-O_2_)TPPCl and (Fe-O_2_)TC_4_PCl compounds in the octet ground spin state were obtained using the ωB97XD/def2-SVP geometry method. The corresponding results are presented in Figure 5 and Appendix A, and the geometry parameters are given in Appendix A. All compounds were optimized using the same methodology. Previous studies have shown that in the absence of a crown ether ring attached to the *meso*-site of the porphyrin ring, all atoms on the porphyrin ring are arranged in a planar configuration [29,42,43,44,45]. The coordination of Fe(III) from FeCl_3_ with the N atom in the porphyrin ring did not significantly affect the planar configuration of the porphyrin ring. In both FeTPPCl and FeTC_4_PCl, the Fe(III) atom was found to be outside the porphyrin plane, and the bond lengths of N-Fe and Fe-Cl were determined to be 2.08 Å and 2.20 Å, respectively. This suggests that the structural features of the Fe(III)-porphyrin ring were not affected by the introduction of the crown ether ring. Analysis of the oxygenated complexes (Fe-O_2_)TPPCl and (Fe-O_2_)TC_4_PCl revealed that the O_2_ molecule effectively induced a change in the conformation of the Fe(III) porphyrin ring. As shown in Appendix A and Appendix A, the presence of the crown ether ring and O_2_ induced a non-planar “saddle-like” conformation of the Fe(III) porphyrin ring in (Fe-O_2_)TC_4_PCl. The crown ether ring and O_2_ had a dual-spatial-site-blocking effect, inducing a more pronounced variation of the Fe(III) porphyrin conformation seen in (Fe-O_2_)TC_4_PCl, which was confirmed by the changes in bond lengths and dihedral angles C_1_-N-N_p_-C_p1_ and C_2_-N-N_p_-C_p2_ in connection with Fe-O and Fe-Cl. However, the bond length of N-Fe was not significantly affected and remained at a value of 2.00 Å. In summary, the crown ether ring induced significant changes in the spatial configuration of the porphyrin ring with Fe(Ⅲ) as the coordination centre but had almost no effect on the interatomic bond length.

### 3.6. Atomic Charge Analysis

An atomic charge is a fundamental and intuitive descriptor of the charge distribution in a given chemical system. However, due to the unobservable nature of the atomic charge and the lack of an objective and unique definition, numerous methods exist to calculate atomic charge [46,47]. The Atomic Dipole Corrected Hirshfeld Atomic Charge (ADCH) method is a more accurate approach for atomic charge analysis. This method is based on the Hirshfeld charge analysis, which defines atomic charge (Equation (1)) as a weighted sum of electron density contributions from neighbouring atoms:(1)$$q_{A}\;=-\int w_{A}(r)∆\rho (r)dr$$
where
$$∆\rho \left(r\right)=\rho \left(r\right)-\sum_{A}\rho_{A}^{0}(r),\;w_{A}(r)=\frac{\rho_{A}^{0}(r)}{\sum_{A}\rho_{A}^{0}(r)}$$
where and $\sum_{A}\rho_{A}^{0}$(*r*) represents the electron density of all atoms in the free state; $∆\rho \left(r\right)$ represents the deformation density, which shows the variation of the electron density during the chirality process after the atoms form molecules. $w_{A}(r)$ is the A-atom weight function, defined as the region in the whole real space belonging to the A-atoms. However, the Hirshfeld charge data are generally small [48], and the dipole moment and electrostatic potential are poorly reproducible [49] mainly because the influence of the atomic dipole moment in the calculation process is neglected. Therefore, Lu Tian [50] proposed the ADCH method, which is a method that defines the atomic dipole moment ($\mu_{A}$) as:(2)$$\mu_{A}\;=\;-\int w_{A}(r)∆\rho (–)(r-r_{A})dr$$

In this method, the Hirshfeld charge of each atom and its $\mu_{A}$ are calculated first, and then each $\mu_{A}$ is expanded into the calibrated positive charge of the surrounding atoms according to Equation (3).
(3)$$\mu_{A}=\sum_{B}∆q_{AB}r_{B}$$

$∆q_{AB}$ denotes the calibrated positive charge of the $\mu_{A}$ of the unfolded A-atom on the B-atom. Finally, after unfolding the $\mu_{A}$ of all atoms into the correctional charge and then accumulating it to the original Hirshfeld charge, the ADCH charge is obtained.

To investigate the influence of various types of *meso*-substituents on the active centres (mainly Fe(III), N_pyrr,_ and N_pyri_) in the Fe(III) porphyrin ring, the four complexes above were analysed using the ADCH method. The results in Table 2 and Appendix A demonstrate that the total number of atomic charges on *meso*-substituents in FeTPPCl and FeTC_4_PCl were identical, with each complex having a total atomic charge of 0.042 a.u and 0.033 a.u, respectively. This suggests that the electron-donating ability of the same type of substituents to the entire Fe(III) porphyrin ring is similar within each complex. Notably, the *meso*-substituents in FeTC_4_PCl exhibited a significantly greater electron-donating property than FeTPPCl, leading to a higher atomic charge of Fe(III) in FeTC_4_PCl and a slightly lower atomic charge regarding the Cl atom. It may explain the superior catalytic activity of FeTC_4_PCl over FeTPPCl.

After analysing the changes in atomic charge in complexes formed by FeTPPCl and FeTC_4_PCl following O_2_ adsorption, it was found that the most significant change in atomic charge occurred in (Fe-O_2_)TC_4_PCl compared to (Fe-O_2_)TPPCl. The data presented in Table 2 indicate that the charge numbers of N_pyrr_ and N_pyri_ were averaged out, while the atomic charge of Fe(III) decreased by 0.029 atomic units (a.u.). This decrease was primarily due to the insertion of O_2_ on top of Fe(III), resulting in non-uniform atomic charge distribution, with the most significant effect being on *meso*-substituents. This effect further decreased the positive charges carried by the four substituents, as depicted in Appendix A. The primary reason for this change was the partial delocalization of electrons in the O_2_ molecule into the Fe(III) porphyrin ring, which was confirmed by the positive charge of 0.045 a.u. in the O_2_ molecule. Furthermore, in (Fe-O_2_)TPPCl, the charge number of the O_2_ molecule was 0.025 a.u., which was smaller than that of (Fe-O_2_)TC_4_PCl, indicating that the crown ether ring substituent further activated the O_2_ molecule and facilitated the catalytic reaction.

### 3.7. Molecular Orbital and Spin Density Analysis

To investigate the influence of the suspension of the crown ether ring by the porphyrin ring on the coordination centre Fe(III) and its bound O_2_, we analysed the electronic interactions of Fe(III)-porphyrin and its corresponding complexes using frontier orbital theory for the highest energy molecular orbital (highest occupied orbital HOMO) and the lowest energy molecular orbital (lowest unoccupied orbital LUMO), respectively. The molecular orbital energy levels of the complexes (Appendix A) showed that in the FeTPPCl and FeTC_4_PCl complex systems, the HOMO–LUMO band gap remained the same, indicating that the introduction of the crown ether ring at the *meso*-position of the porphyrin ring did not affect the electron density distribution in the orbitals, which was mainly concentrated in the porphyrin ring and its corresponding Fe(III) ion.

Upon further interaction of the FeTPPCl and FeTC_4_PCl complexes with molecular oxygen to form dioxygen complexes ((Fe-O_2_)TPPCl and (Fe-O_2_)TC_4_PCl), we observed that the HOMO–LUMO band gap corresponding to the (Fe-O_2_)TPPCl complex that was formed after FeTPPCl interacted with O_2_ did not change significantly, as seen in Appendix Aa,c. However, the dioxygen complex (Fe-O_2_)TC_4_PCl formed after the interaction of FeTC_4_PCl with O_2_ exhibited a significant reduction in the HOMO–LUMO band gap, particularly in the HOMO and LUMO orbital energy of the β molecular orbital, as observed in Figure 6. This may be attributed to the electron-donating ability of the crown ether ring, which increases the electron density of the Fe(III) ion, thereby enhancing its binding and electron transfer ability in connection with O_2_ and ultimately improving its catalytic activity.

In summary, our results demonstrate that introducing the crown ether ring at the *meso*-position of the porphyrin ring can significantly enhance the catalytic activity of Fe(III) complexes regarding O_2_, which may have important implications for the development of novel catalysts for various chemical reactions.

To further illustrate the distribution of α and β electron densities in the four open-shell aforementioned complexes, and to examine the distribution of unpaired electrons in three-dimensional space, this study employed the Multiwfn software to conduct spin density analysis. Figure 7 presents the three-dimensional spatial isosurface after full-space integration, highlighting the difference between the electron density of α orbitals (ρ_α_) and β orbitals (ρ_β_), which is indicated by green and blue colours, respectively. A positive value in green indicates a higher number of α orbital electrons than β, while a negative value in blue indicates the opposite.

Based on Figure 7, all four complexes show a higher concentration of α electrons than β electrons, which is consistent with the previous computational results. Further examination reveals that in FeTPPCl and FeTC_4_PCl, α electrons are mainly located on Fe(III), Cl, N_pyrr_, and N_pyri_ atoms. Upon interaction with O_2_, α electrons transfer to the α-C atoms of pyridine and pyrrole rings. Of particular note is that the introduction of the *meso*-substituted crown ether ring in (Fe-O_2_)TC_4_PCl affects the electron distribution of the Fe(III) porphyrin ring, changes the HOMO orbital energy, activates molecules, and reduces the HOMO–LUMO energy difference (as shown in Figure 6). However, this modification also results in an increased instability of the entire molecular system. Therefore, further discussion is required to explore this research aspect.

### 3.8. Density-of-State Analysis

The density-of-state (DOS) diagram depicts the distribution of energy levels for molecular orbitals (MOs) in a chemical system. The DOS curve reflects the number of MOs within a given energy level in a unit energy interval [36,51,52,53]. Figure 8 and Appendix A exhibit the TDOS and PDOS diagrams and maps for FeTPPCl, FeTC_4_PCl, (Fe-O_2_)TPPCl, and (Fe-O_2_)TC_4_PCl, along with the contributions from different MO groups. Since the four complex systems have an open-shell configuration, α and β MOs were plotted separately in the DOS and PDOS curves. The upper half of the graph frame displays the curves of α MOs as solid lines, while the lower half shows the curves of β MOs as dotted lines. Symmetric α and β MOs signify that electrons in both orbitals have no spin hybridization in the same energy range, although spin polarization arises otherwise.

The changes observed in Figure 8 and Appendix A demonstrate that the TDOS curves of α and β MOs for each complex are nearly symmetrical, indicating almost equal spin polarization of electronic states in α and β orbitals as well as a uniform contribution of molecular orbitals to TDOS. However, the contribution of each group to TDOS varies. To further analyse the impact of different groups on TDOS in various complexes, PDOS was used to examine the Fe–Cl bond, pyridine N (N_pyri_) on the porphyrin ring, substituted groups on the *meso*-position, and the angular momentum of S, P, and D in the complex. As shown in Figure 8, P orbitals contribute significantly to the entire orbital energy level in all complexes, particularly regarding the molecular frontier orbitals of HOMO and LUMO. On the other hand, S orbitals only contribute to TDOS in the lower energy level.

Additionally, the spherically symmetric distribution of S orbitals in Appendix A suggests the symmetric spin MOs of α and β. By analysing the contribution of d orbitals to α and β MOs, it was found that the contribution of d orbitals to TDOS is minimal, but that the evident spin polarization occurring in α and β MOs is primarily due to Fe(III) being the only provider of d orbitals occupied by electrons in all systems. Among all the contributions of angular momentum to TDOS, the proportion of d orbitals is the smallest. The spin multiplicity of all studied systems was set to sextet states for FeTPPCl and FeTC_4_PCl complexes and octet for (Fe-O_2_)TPPCl and (Fe-O_2_)TC_4_PCl, resulting in different electron degeneracies on α and β MOs.

The contribution of porphyrin N_total_ (the total of the N_pyri_ and N_pyrr)_ and *meso*-substituent (*meso*-) to the total density of states (TDOS) in Fe–Cl and porphyrin rings shown in Figure 8 was analysed, which revealed that while these groups contribute to the orbital energy levels throughout the entire interval, their overall contribution is not significant. A further comparison revealed that the *meso*-crown ether ring-substituted group contributed most significantly to TDOS (Figure 8 and Appendix A). The (Fe-O_2_)TC_4_PCl complex was especially analysed, and the difference in highest occupied molecular orbital (HOMO) between α and β MOs was found to be 1.89 eV, which can be determined by analysing the HOMO difference of α and β MOs in FeTPPCl (0.13 eV), which indicates that the introduction of the crown ether ring primarily caused this change. Combined with the analysis in Section 3.6, it can be seen that the crown ether ring helps to influence the coordination field environment of the FeTC_4_PCl complex binding O_2_, alter the electron spin polarization of Fe(III), and improve its catalytic activity.

### 3.9. Thermodynamic Quantities Analysis

The study of thermodynamic quantities involves electron energy, vibration energy, and geometry concerning the structure and other information in quantum calculations for the electron energy obtained under a specific geometry. The study of practical problems needs to be more accurate. Gibbs free energy (G), internal energy (U), entropy (S), enthalpy (H), and other thermodynamic quantities such as those that are a function of temperature, and the temperature setting affects their values [3,38]; therefore, in this paper, the thermodynamic changes of the complexes during the reaction as well as their differences before and after the reaction were studied in depth using Shermo software. At the same time, the changes in thermodynamic quantities at different temperatures were calculated.

Table 3 summarizes the electronic energy [E_ele_], internal energy [U], enthalpy [H], and free energy [G] of each complex at a steady state, and Figure 9 plots the corresponding thermodynamic amounts with temperature regarding the thermal corrections to internal energy (U_corr_), enthalpy (H_corr_), and Gibbs free energy (G_corr_). It can be seen from Table 3 that when the FeTPPCl and FeTC_4_PCl complexes combine with O_2_ molecules to form oxygenated complexes ((Fe-O_2_)TPPCl and (Fe-O_2_)TC_4_PCl), their corresponding thermodynamic quantities significantly decrease, especially ΔG < 0, indicating that the two complexes can find adsorption with O_2_ and form lower-energy oxygenated complexes. However, it was found from the previous experimental results that an increase in reaction temperature has a more significant enhancement on the reaction rate and product selectivity. To explore the reason for this, the process of G, U, and H was analysed using Shermo software. It can be seen from Figure 9 that when the temperature was further increased, the change of thermodynamic quantities (E_ele_, U, H, and G) was found to be nonlinear with increasing temperature. The trend of the G curve was more extensive than that of the U and H curves (see Figure 9a), and the trend of the curve of G_corr_ was also more extensive than that of U_corr_ and H_corr_, which shows that the effect of temperature on this reaction cannot be neglected.

### 3.10. Discussion on the Mechanism of the Oxidation of Cyclohexene

A proposed reaction mechanism is presented in this study to investigate the oxidation process of cyclohexene with a carbon–carbon double bond using FeTC_4_PCl and O_2_ as oxidants at standard atmospheric pressure without any reducing agent and additives. By combining experimental data, theoretical analysis, and product analysis, it is concluded that the oxidation of cyclohexene by O_2_ occurs mainly in the allylic position. Previous research on this reaction mechanism (particularly the study by Mohebi et al. [54]) was considered in proposing the mechanism presented in Figure 2. The proposed mechanism includes multiple steps [14,55,56,57], with the primary oxidation process being a free radical chain reaction. Initially, the catalyst FeTC_4_PCl was activated at a specific temperature, as observed in Figure 2 and Figure 3. The low conversion rate of the substrate within 1 h of the reaction verifies this activation process. Fe(III) in FeTC_4_PCl and O_2_ undergo surface adsorption at a specific temperature to form a compound containing O_2_ molecules ((Fe-O_2_)TC_4_PCl). This compound can combine with protons in the system to form an unstable product ((Fe^IV^-OOH)TC_4_PCl), which then loses H_2_O molecules to form a catalytically active high-valence iron intermediate [(Fe^IV^ = O)TC_4_PCl]^+•^. This high-valence complex can interact with the carbon–carbon double bond with cyclohexene, leading to the removal of a proton from the allylic carbon and the generation of a relatively stable allyl radical and a high-valence intermediate ((Fe^IV^-O-H)TC_4_PCl (Step 1)). This step completes the initiation process of the cyclohexene radical, which is considered the chain initiation of free radicals. After the formation of the allyl radical, it rapidly reacts with O_2_ molecules in the system to generate an allyl peroxygen radical (step 2). It then undergoes a radical reaction with the substrate cyclohexene molecule (step 3). These reactions produce an allyl radical and product **3** or generate the transition-state product of 2-cyclohexyl-1-yl-hydrogen peroxide. The allyl free radical generated can either continue the process triggered by the free radical chain or undergo subsequent reactions. The 2-cyclohexyl-1-yl-hydrogen peroxide produced can react with FeTC_4_PCl to generate (Fe^IV^-O-H)TC_4_PCl and an alkoxy free radical (step 4). The alkoxy radical can react quickly with cyclohexene to form product 1 (Step 5), which is then further oxidized at higher temperatures to form product 2 (Step 10). During step 5, the substrate cyclohexene loses the protons on the allylic carbon to form an allyl radical, part of which comes from the previous step. These allyl radicals can combine hydrogen and oxygen to oxidize the active 2-cyclohexen-1-yl-hydroperoxide (step 6) in the oxidation system. In the presence of the active species (Fe^IV^-O-H)TC_4_PCl, a homolytic reaction occurs by forming the alkoxy radical and the hydroxide radical, which are the main products (Step 7). Subsequently, the alkoxy radical reacts with the previously generated 2-cyclohexen-1-yl-hydroperoxide, producing product **2** and a new alkoxy radical (Step 8). The newly formed alkoxy radical undergoes a propagation reaction similar to the reactions described in Steps 5 and 8. Some may combine it with the hydroxyl radical to yield products **1** and **2** (Step 9). The concentration of the alkoxy radical decreases due to the formation of products **1** and **2**, resulting in a decrease in the oxidation reaction rate. This mechanism demonstrates the crucial role played by FeTC_4_PCl in initiating and propagating radicals. Additionally, the catalytically active Fe(III) centre may promote the change in its valency [42], facilitating the formation of the active intermediate in the catalytic oxidation reaction.

## 4. Conclusions

This paper investigates the impact of *meso*-position substituents on the catalytic activity of Fe(III) porphyrin complexes as well as examines the catalytic oxidation of cyclohexene under mild and without reductant conditions. The results show that *meso*-substituents effectively influence the catalytic activity of Fe(III) and shorten the reaction induction period. FeTC_4_PCl with *meso*-5,10,15,20-*tetra*-(15-crown-5-benzo-3-yl-sulfonyloxophen-4-yl) substituents exhibit higher conversion rates and selectivity compared to FeTPPCl with *meso*-phenylhydroxy substituents. Moreover, introducing pyridine ligands inhibits the oxidation reaction by affecting the redox potential of the active metal centre.

Furthermore, the paper employs the DFT method to optimize the geometric structure, molecular orbital energy level analysis, atomic charge, spin density, and density of orbital states of FeTPPCl, FeTC_4_PCl, and the oxygenated complexes formed after the adsorption of O_2_. The (Fe-O_2_)TCPPCl and (Fe-O_2_)TC_4_PCl oxygenated complexes exhibit distinct characteristics, with the planar structure of Fe(III)-porphyrin changing to a nonplanar “saddle-like” conformation in (Fe-O_2_)TC_4_PCl. The crown ether ring substituent impacts the electron distribution of the Fe(III) porphyrin ring, modifies the HOMO orbital energy, activates the molecule, and further reduces the energy level difference of HOMO–LUMO, indicating its auxiliary activation effect on the catalytic system composed of the Fe(III) porphyrin ring. However, additional research is necessary to fully comprehend the intricacies of these changes.

In summary, this study provides new insights into the design of *meso*-substituted Fe(III) porphyrin catalysts for catalytic oxidation reactions. Based on experimental and theoretical analyses, this paper deduced the mechanism of the cyclohexene oxidation reaction using FeTC_4_PCl as the catalyst and O_2_ as the oxidant. The reaction proceeds by generating the high-valent iron intermediate [(Fe^IV^ = O)TC_4_PCl]^+^ initially, followed by the formation of (Fe^IV^-O-H)TC_4_PCl and an alkoxy radical transition state. Radical transfer and transfer subsequently generate the main three products. The study also reveals that the reaction mechanism follows a free radical chain reaction process, and the crown ether ring affects the activity of the catalytic centre of Fe(III) and demonstrates the property of shortening the reaction induction period. These findings provide new theoretical insights into modifying metalloporphyrins with *meso*-substituent groups.

## Data Availability

The data that support the findings of this study are available from the corresponding author upon reasonable requests.

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
