# Peer review of "Experimental and Theoretical Study on Crown Ether-Appended-Fe(III) Porphyrin Complexes and Catalytic Oxidation Cyclohexene with O_2"

_molecules, 2023, doi:10.3390/molecules28083452_

Round 1

Reviewer 1 Report

In this submitted manuscript (molecules-2315247), Dr. Feng and coworkers reported their studies in both experimental and theoretical aspects on a crown ether functionalized Fe(III) porphyrin complex and its effects on cyclohexene oxidation with oxygen. It is important to investigate the effectiveness of different Fe(III) porphyrin derivatives on alkene oxidation reactions, and the interesting results observed here would appeal to the broad readership of Molecules. However, clearer descriptions/discussions and extra experiments are recommended for this manuscript before further consideration.

Major criticisms include:

1.    It is interesting to see the authors could measure and calculate how much oxygen is consumed by gas chromatography on lines 82-83. Could the authors explain more about how was this done?

2.    More entries for finding the optimal reaction temperature are recommended. In Table 1, the authors claimed that 70oC is the optimal reaction temperature because the entry with 77oC gives conversion dropped by 3%. Two more temperatures are recommended to be measured at 90oC and 100oC. It is not safe to determine the summit with only one data dropped down. With at least two to three sliding conversion data, we can confidently say higher temperatures passing 70oC would result in side effects on the starting material conversion.

3.    In section 3.2, the authors mentioned that “In comparison, FeTPPCl as the catalyst showed moderate increases in cyclohexene conversions” on lines 154-155. However, following that sentence, the authors wrote “indicating the catalytic activity of FeTC4PCl was significantly higher than that of FeTPPCl” on lines 156-157. It is confusing to read this as these two sentences draw two completely different and conflicting conclusions. The authors should clarify it.

4.    In Table 1, what are the differences between entries 5 and 9? From what the authors provided, it looks like these two entries have the same reaction conditions (reagent, catalyst, and temperature). Why the results are different, in conversion, TON, and product selectivity?

5.    It is recommended that the authors should mention if the data in Table 1 is from a one-time reaction, or from the average of several parallel experiments. The same for data in Table 2, Figures 2 and 3.

6.    There is no need to duplicate the same data in two Tables. It is evident that the data in Table 2 have been discussed completely in Table 1, and in this case, Table 2 can be removed from the manuscript.

7.    The language and format in some places need to be improved/revised. For example, “tetra-” on line 45 should be italicized; DFT should be in a parenthesis on line 61; no need to capitalize the first letter of “Density” on line 65; in section 3.2, some of the numbers are bold while others are not, like “Fig. 2 and 3” on line 152; the sentence on line 222 “significantly affected the conversion and oxidation products” is recommended to be revised as “significantly affected the conversion and the selectivity of the oxidation products”; the symbol on line 272 “∆ρ(r)” is different from the one used in equation 1; the name of the author should be capitalized with the first letter for “Lu tian” on line 278; the space is missed between the word and symbol “theµAof” on line 285.

Reviewer 2 Report

The work reported in this manuscript is interesting and well presented. However, there should be further improvement and revision before the acceptance. Some comments are:

1.     In Figure 8, β-HOMO is missing.

2.     Reaction transition state should been calculated.

3.     The discussion is poor, please improve.

4.     Some articles must be cited in the paperApplied Surface Science, 2022, 589153002.Fuel, 2023, 342, 127890.; Chemosphere, 2023, 317, 137823; J Mater. Sci-Mater El. 32 (8) (2021) 10277-10288.

Round 2

Reviewer 1 Report

In the revised manuscript (molecules-2315247-v2), the authors have discussed the questions raised by the reviewers and provided sufficient explanations for them. Proper changes have been made in the manuscript to improve its quality to meet the requirements for publication. Based on that, I'm happy to recommend accepting it in its current version.